# The Association Between Breakfast Skipping and Positive and Negative Emotional Wellbeing Outcomes for Children and Adolescents in South Australia

**DOI:** 10.3390/nu17081304

**Published:** 2025-04-09

**Authors:** Sophie Burnell, Mary E. Brushe, Neida Sechague Monroy, Tess Gregory, Alanna Sincovich

**Affiliations:** 1School of Public Health, University of Adelaide, Level 4 Rundle Mall Plaza 50 Rundle Mall, Adelaide, SA 5000, Australia; sophie.burnell@sa.gov.au (S.B.); mary.brushe@thekids.org.au (M.E.B.); tess.gregory@thekids.org.au (T.G.); 2The Kids Research Institute Australia, The University of Western Australia, 108 North Terrace, Adelaide, SA 5000, Australia; nsechagu@umich.edu; 3UWA Centre for Child Health Research, The University of Western Australia, 35 Stirling Highway, Crawley, WA 6009, Australia

**Keywords:** breakfast consumption, emotional wellbeing, school-aged children, Wellbeing Engagement Collection, life satisfaction, worries

## Abstract

**Background/Objectives**: The prevalence of child and adolescent breakfast skipping is concerning, and limited existing evidence suggests an association between skipping breakfast and negative emotional wellbeing outcomes. However, positive emotional wellbeing outcomes have been neglected from research in this space. **Methods**: This study explored child and adolescent breakfast skipping and associations with both positive and negative emotional wellbeing outcomes. We utilised existing population-level data (*n* = 80,610, aged 8–18 years) collected in 2023 via a statewide census among children and adolescents in South Australian schools, the Wellbeing and Engagement Collection. **Results**: Adjusted linear regression analyses indicated lower scores on positive wellbeing outcomes for students who skipped breakfast every day compared to students who never skipped breakfast, ranging from β = −0.19 for happiness (95% CI = −0.21, −0.17) to β = −0.23 for optimism (95% CI = −0.25, −0.21). Results also highlighted higher scores on negative wellbeing indicators, sadness (β = 0.12, 95% CI = 0.10, 0.15), and worries (β = 0.05 95% CI = 0.03, 0.08) among students who always skipped relative to those who never skipped breakfast. **Conclusions**: Findings support the potential for child and adolescent emotional wellbeing to be fostered through interventions designed to promote daily breakfast consumption. Future research focused on gaining a deeper understanding of the circumstances surrounding child and adolescent breakfast consumption behaviours is needed to inform the development of effective interventions to increase breakfast consumption.

## 1. Introduction

### 1.1. Breakfast Consumption in Childhood and Adolescence

Widely referred to as the most important meal of the day, breakfast provides an opportunity to replenish energy stores and contribute to daily vitamin and mineral consumption [1]. For children and adolescents, breakfast consumption not only provides nutritional benefits, but is also associated with a variety of health [2,3], behaviour [3,4], education [3,4,5,6,7], and emotional wellbeing outcomes [3]. Concerningly, the World Health Organization (WHO) has observed overall declines in breakfast consumption between 2014 and 2018 across 45 countries [8]. In 2017–2018 surveys, on average 42% of adolescents aged 11–15 years reported not eating breakfast every school day, with this figure at 38% in 2009–2010 [8,9,10]. Similarly, in 2021, Australian evidence indicated that 44.9% of students aged 8–18 years skipped breakfast at least once per week [11].

Skipping breakfast is associated with other unhealthy behaviours in adolescents, including smartphone overuse, smoking, alcohol overuse, and physical underactivity [3,4]. Breakfast consumption has also been associated with a decreased likelihood of experiencing sleep difficulties in students aged 11–15 years across 42 countries [2]. Limited research has focused on the association between breakfast consumption and child mental health and wellbeing, albeit with the sparse research to date focused on negative measures of mental health and wellbeing (e.g., loneliness, anxiety, depression) [12,13,14,15,16], and little to no mention of positive functioning measures (e.g., optimism, happiness, life satisfaction) [17].

### 1.2. Breakfast Consumption and Mental Health

Childhood and adolescence are important stages in which physical, emotional, and cognitive changes occur. Good health (physical and mental) and optimal nutrition have been identified as one of five necessary domains for adolescent wellbeing as part of the WHO’s Global Accelerated Action for the Health of Adolescents [18]. Poor mental health is a widespread issue for young people [19,20] and a top five cause of morbidity (ages 10–19) and mortality (ages 15–19) globally [18,21]. Data among 275,057 adolescents aged 12–17 years across 82 countries, collected from 2003 to 2015, highlighted a pooled prevalence of 14.0% for suicidal ideation (95% CI: 10.0, 17.0) and 9.0% for anxiety (95% CI: 7.0, 12.0%) [20]. In 2019, anxiety and depressive disorders resulted in 1.7 million years lost to disability amongst children aged 5–14 years worldwide [22].

As the burden of poor mental health intensifies for children and adolescents [23], it is important to prioritize investments in supportive and preventative measures. The potential influence, positive or negative, of breakfast consumption on child and adolescent mental health and wellbeing is yet to be adequately understood. The risk of poor emotional wellbeing (e.g., feeling worthless, confused, angry, stressed, having a depressive mood) has been shown to be higher amongst children and adolescents who skip breakfast than their peers who do not [13,17,24]. During the COVID-19 pandemic, Canadian adolescents (*n* = 40,521) who reported increased levels of loneliness had greater odds of skipping breakfast, and this was seen for both boys (OR 1.40, 95% CI: 1.32, 1.49) and girls (OR 1.62, 95% CI: 1.53, 1.71) [25]. Having pain or discomfort and feeling worried, sad, or unhappy was reported more frequently for students who never or almost never ate breakfast, compared to those who ate breakfast every day or often (*n* = 783, aged 11–16 years), across four schools in China [24]. Among a sample of 14,880 Iranian students aged 6–18 years, those who skipped breakfast two or more days a week had higher odds of psychological distress (e.g., worthlessness OR 1.77, 95% CI: 1.53, 2.04; angriness OR 1.67, 95% CI: 1.50, 1.86), compared to students who consumed breakfast at least five days per week [13]. An online youth survey of Korean students (*n* = 65,528, mean age 15 years) found higher odds of perceived stress (OR 1.23, 95% CI 1.18, 1.27) and depressive mood (OR 1.22, 95% CI 1.18, 1.27), as well as lower odds of general health (OR 0.86, 95% CI 0.82, 0.89) and happiness (OR 0.80, 95% CI 0.77, 0.83), when five or more days of breakfast were missed [17]. Another study, sampled from the same survey (*n* = 62,276, mean age 15 years), demonstrated that a depressive mood (26.68% versus 23.45%), very high stress levels (12.83% versus 9.14%), and suicidal ideation (13.83% versus 11.30%) were more likely to be reported among students who skipped breakfast (i.e., ate breakfast <2 days per week) compared to children who did not skip (i.e., ate breakfast at least five days a week) [12].

Overall, the existing literature indicates poorer mental health and wellbeing outcomes (e.g., depression, loneliness, and stress) among children and adolescents who skip breakfast. Further research should consider the relationship between breakfast skipping and positive mental health and wellbeing outcomes (e.g., happiness and life satisfaction) as they represent the strengths of individuals and provide valuable insight into leverage points to promote and protect child and adolescent mental health and wellbeing [26,27].

### 1.3. Current Study

Previous research has shown that regular breakfast consumption has potential as a modifiable health behaviour for the promotion of a number of health, behaviour, and educational outcomes [7,17,24]. While limited research has investigated the link between breakfast and mental health [12,13], this study extends the investigation of breakfast skipping and emotional wellbeing outcomes in children and adolescents to be inclusive of both positive and negative aspects of emotional wellbeing. Furthermore, this study positions children’s voice as critical to the research by allowing students to self-report their own feelings about their wellbeing without relying on others to make assumptions about their mental health, Finaly, by using data captured at a population level, the findings can be generalized to a greater diversity of students and better inform public health efforts to support student wellbeing. By addressing these gaps in the field, the study will allow for a clearer understanding of the relationship between breakfast consumption and mental health and wellbeing, which will help strengthen the current evidence base regarding the benefits of approaches designed to promote breakfast consumption among children and adolescents.

The current study was designed to answer the following question: Is skipping breakfast associated with higher levels of negative emotional wellbeing (sadness and worries) and lower levels of positive emotional wellbeing (happiness, life satisfaction, optimism) for South Australian children and adolescents?

## 2. Materials and Methods

### 2.1. Data Source

The South Australian Department for Education (SA DfE) conducts the Wellbeing and Engagement Collection (WEC) annually, with all schools in South Australia invited to take part [28,29]. Students in Grades 4 to 12 are invited to participate and complete the WEC during school hours via an online data collection system, usually taking between 25 and 45 min. Student WEC responses are collected using both single-item and multi-item questions across four domains (Emotional Wellbeing, Engagement with School, Learning Readiness, and Health and Wellbeing Outside School). This study utilized data from students in government (i.e., public) schools who participated in the 2023 WEC.

Student-level demographic data (e.g., language background, parental education, geographical remoteness) were sourced from the SA DfE via school enrolment records. The SA DfE linked the student level demographic data and WEC response using each child’s unique education ID number, before providing data to researchers for analysis.

### 2.2. Participants

Figure 1 presents a flow chart identifying the sample population for this study. A total of 122,177 children were enrolled in grades 4 to 12 in government schools within South Australia and thus eligible to participate in the 2023 WEC. Students who did not participate in the 2023 WEC and did not have valid WEC data (*n* = 35,546), those with missing data on the question about skipping breakfast (*n* = 4576), or those missing data on all five emotional wellbeing items (*n* = 390) were excluded from the study population. Imputation analysis was conducted for students with missing data on at least one emotional wellbeing item (*n* = 942) or missing data on at least one confounding variable (*n* = 5539). The total imputed sample used in this study comprised 80,610 students, representing 66.6% of the eligible sample. Appendix A Table A1 presents the sociodemographic characteristics of the eligible sample relative to the imputed sample.

### 2.3. Measures

#### 2.3.1. Exposure

The frequency of student breakfast consumption was measured in the WEC by the question ‘How often do you eat breakfast?’, representing a typical weekly (or habitual) intake and offering eight response options reflecting frequencies in the range “Never”, “1 day per week” …, “Everyday” [28]. These data were recoded to simplify breakfast habits into four categories, reflecting never (skip 0 days), sometimes (skip 1–3 days), often (skip 4–6 days), and always (skip 7 days) breakfast skippers, in alignment with previous research [11].

#### 2.3.2. Outcomes

##### Positive Emotional Wellbeing

Measures of positive emotional wellbeing utilized include student’s self-reported happiness, life satisfaction, and optimism, reflecting the ability of students to manage emotions and effectively navigate everyday life [21]. Life satisfaction was measured using a 5-item Satisfaction with Life Scale adapted for children [30]. Students were asked their agreement level (“Strongly disagree = 1”, “Disagree” = 2, “Don’t agree or disagree” = 3, “Agree” = 4, and “Strongly Agree” = 5) to the following statements: “In most ways my life is close to the way I want it to be”, “The things in my life are excellent”, “I am happy with my life”, “So far I have gotten the important things I want in life”, “If I could live my life over again, I would have it the same way”. Optimism was measured with a 3-item Optimism scale [31]. Students were asked their agreement level (“Strongly disagree = 1”, “Disagree” = 2, “Don’t agree or disagree” = 3, “Agree” = 4, and “Strongly Agree” = 5) to the following statements: “I have more good times than bad”, “I believe more good things than bad things will happen to me”, “I start most days thinking I will have a good day”. A 4-item Happiness scale was used [32]. Students were asked their agreement level (“None of the time = 1”, “A little of the time” = 2, “Some of the time” = 3, “Most of the time” = 4, and “All of the time” = 5) to the following statements: “I feel happy”, “I have a lot of fun”, “I love life”, “I am a cheerful person”. Higher scores on the positive emotional wellbeing outcomes indicate higher life satisfaction, optimism, and happiness.

##### Negative Emotional Wellbeing

Negative emotional wellbeing was examined through self-reported measures of sadness and worries, which represent a reflection of current student experiences of mental ill health. Sadness was measured using a 3-item scale adapted from the Seattle Personality Questionnaire for young school-aged children [33]. Students were asked their agreement level (“Strongly disagree = 1”, “Disagree” = 2, “Don’t agree or disagree” = 3, “Agree” = 4, and “Strongly Agree” = 5) with the following statements: “I feel unhappy a lot of the time”, “I feel upset about things”, “I feel like I do things wrong a lot”. Worries had a 4-item scale [34]. Students were asked their agreement level (“Strongly disagree = 1”, “Disagree” = 2, “Don’t agree or disagree” = 3, “Agree” = 4, and “Strongly Agree” = 5) to the following statements: “I worry a lot about things at home”, “I worry a lot about things at school”, “I worry a lot about mistakes I make”, “I worry about things”. Higher scores on the negative emotional wellbeing outcomes indicate higher levels of sadness and worries.

#### 2.3.3. Confounding

Student age, gender, community-level socioeconomic status, language background, parental education, geographical remoteness, self-reported overall health, and sleep were included as confounding factors (see Figure 2 for a directed acyclic graph representing the theoretical relationship between breakfast skipping and emotional wellbeing). Previous research demonstrates an increased prevalence of regularly skipping breakfast among girls, older students, and students with greater socioeconomic disadvantage [3,8,9,10,11]. Age and gender can also influence emotional wellbeing, as these factors shape social and community norms [35]. Additionally, childhood health and behaviours are known to be influenced by parental education and socioeconomic status [35].

Gender included self-reported response options, “Male”, “Female”, and “Other”, with school enrolment data used where gender data were missing. Age values were sourced from the 2023 school enrolment census. A categorical age variable was generated representing the closest whole year value for each child, ranging from 7 to 17 years. Overall health was measured by the single WEC item ‘In general, how would you describe your health?’ with four responses (“Poor” = 1, “Fair” = 2, “Good” = 3, “Excellent” = 4). Sleep was measured with the question “How often do you get a good night’s sleep?”; students were offered eight response options, reflecting frequencies from “Never” and “1 day per week” to “Everyday”. Linked data from students’ school enrolment censuses were also used to obtain the remaining confounding variables. Language background was divided by primary language (“English only” = 0, “Non-English speaking background” = 1). Socioeconomic position was reported as per the Socio-Economic Indexes for Areas Index of Relative Socio-economic Advantage and Disadvantage, derived from Australian Bureau of Statistics census information (“Most disadvantaged” = 1, …, “Most advantaged” = 5) [36]. Parental education reflected the highest level of education received by a parent/primary caregiver; responses were recoded into categories (“Year 12 or below” = 1, “Certificate” = 2, “Diploma” = 3, and “Bachelor’s degree and above” = 4). Geographical remoteness was given as per the Accessibility and Remoteness Index of Australia and recoded into the following categories: “Major cities of Australia” = 1, “Inner regional Australia” = 2, “Outer regional Australia” = 3, and “Remote/Very remote Australia” = 4 [37]. The remote and very remote categories were combined before analysis due to the low numbers in very remote populations, maintaining participant privacy.

### 2.4. Statistical Analysis

In the analysis sample (*n* = 80,610), multiple imputation was performed to address missing data on outcomes and confounder variables. In this sample, 942 (0.8%) students had missing data on one to four emotional wellbeing outcomes, ranging from 228 (0.3%) for optimism to 510 (0.6%) for happiness. Additionally, 5539 (4.6%) had missing data on at least one confounder variable, ranging from 2 (0.0%) for age to 2434 (3.0%) for the highest parental education level. All analysis variables, along with auxiliary variables, were included in the imputation model to improve the prediction of missing data in the study variables. The auxiliary variables included measures from the 2022 WEC (i.e., life satisfaction, optimism, happiness, sadness, worries, highest parental education level, overall self-rated health, and good night sleep) and measures from the current collection year (2023) (i.e., emotional regulation, school belonging, peer belonging, highest level of parental occupation, and participation in school card scheme eligible for low-income families). Multiple imputation was conducted using the mi impute chained command with 30 imputed datasets and a burn-in of 30 iterations. The results of the imputed analysis (*n* = 80,610) were similar to the complete case analysis (*n* = 74,129) and thus imputed results are presented in the main text.

Descriptive statistics were explored for confounding and emotional wellbeing variables, separated by breakfast skipping category. A series of linear regressions were used to assess associations between breakfast skipping and each emotional wellbeing outcome of interest, and these tests were run before and after adjustment for confounding variables (age, gender, community socioeconomic status, language background other than English, parent education, level of remoteness, student’s overall health and student’s sleep quality). Students who never skipped breakfast were used as the reference group. The regression analyses were conducted using the regress command with vce (cluster). The cluster command was used to account for the nested nature of the data at the school level. The linear regression output is presented as a beta-coefficient value with a 95% confidence interval. All analyses were generated with STATA 17 software [38].

## 3. Results

### 3.1. Sociodemographic Characteristics

The sociodemographic characteristics of students included in this study are presented in Table 1, separated by breakfast skipping frequency. Notably, students who always skip breakfast were two years older on average. Students who always skip breakfast had a mean age of 14 years compared to never skippers, who had a mean age of 12 years. In terms of gender, 43.3% of never breakfast skippers were female, compared to 58.4% of always skipping female students. Of the male respondents, never skipping breakfast (56.7%) was reported more than always skipping breakfast (41.6%). A higher percentage of students who never skip breakfast have medium and excellent levels of self-rated overall health (47.2% and 39.5%, respectively). By contrast, half of students that always skip breakfast (48.1%) report poor or fair self-rated overall health. Experiencing good sleep always or often occurred for 79.3% of never breakfast skippers, and 31.9% of students who always skipped breakfast. Students who never skip breakfast were more likely to have a parent with a bachelor’s degree or above (45.8%), compared to students who always skip breakfast (23.4%).

### 3.2. Emotional Wellbeing

The five wellbeing outcomes in this study utilized scales ranging from 1 to 5; the mean scores for these variables are presented in Table 2. For the positive measures of emotional wellbeing (happiness, life satisfaction, and optimism), the mean for the never skipping breakfast students was above the total sample mean, whilst that of sometimes and always skippers was below the total sample means. The inverse was true for the negative measures of emotional wellbeing (sadness and worries), where students who never skip breakfast had a mean score below the total sample, and sometimes and always breakfast skippers had a mean score above the overall mean for all students.

### 3.3. Linear Regression

Table 3 presents linear regression results for the association between breakfast skipping categories and the five emotional wellbeing outcomes. A visual representation of the adjusted associations between breakfast skipping and all five wellbeing outcomes is provided in Figure 3. Compared to students who never skip breakfast, always breakfast skippers reported higher levels of sadness (β = 0.12, 95% CI = 0.10, 0.15) and worries (β = 0.05, 95% CI = 0.03, 0.08). Always skippers had lower positive emotional wellbeing levels compared to always skippers ranging from β = −0.19 for happiness (95% CI = −0.21, −0.17) to β = −0.23 for optimism (95% CI = −0.25, −0.21). Overall, the adjusted model demonstrated stronger associations between breakfast skipping and positive emotional wellbeing outcomes, compared to the relationship with negative emotional wellbeing outcomes.

## 4. Discussion

The objective of this study was to investigate whether skipping breakfast was associated with higher levels of negative emotional wellbeing (sadness and worries) and lower levels of positive emotional wellbeing (happiness, life satisfaction, optimism) among South Australian school-aged children and adolescents. In the current study, students who skipped breakfast (sometimes, often or always) made up 50.8% of the study population. This is a larger percentage than expected given the existing international literature, which has reported that approximately 40% of students skip breakfast on at least one school day [8,9,10]. However, the estimates from previous studies vary based on children’s age and the time period in which breakfast skipping was explored (e.g., across the whole week, compared to only weekdays).

Results demonstrated lower levels of positive wellbeing and higher levels of negative wellbeing for students who always skipped breakfast when compared to those who never skipped breakfast, after adjustment for a comprehensive set of child and family level confounding variables. These findings add to the limited existing evidence that supports the notion that better emotional wellbeing outcomes are associated with regular breakfast consumption. Previous research found breakfast skippers in Iran (*n* = 14,880, aged 6–18 years) to be at an increased risk of poor emotional wellbeing in areas of feeling worthless, angry, worried, confused, depressed, and anxious [13]. Of all of the emotional wellbeing outcomes in this study, the highest increased risk for breakfast skippers was seen for worthlessness (OR 1.77, CI 95% 1.53, 2.04), confusion (OR 1.68, CI 95% 1.43, 1.97), and feeling angry (OR 1.67, CI 95% 1.50, 1.86). Experimental research in the UK (*n* = 213, aged 6–10 years) exploring impacts of cereal consumption for breakfast versus no breakfast, found lower emotional distress, depression, and negative moods among children who consumed cereal [39](Smith, 2010). Research conducted in Spain (*n* = 527, aged 12–17 years) explored associations among breakfast skipping, poor quality breakfast, and good quality breakfast with health-related quality of life, stress, and depression [40]. Interestingly, breakfast skippers reported higher health-related quality of life, with lower stress and depression, compared to adolescents consuming a very poor-quality breakfast (did not eat bread/toast, cereal or dairy products for breakfast, but did eat commercially baked goods). By comparison, this study found that students who consumed a good quality breakfast (ate bread/toast/cereal and/or dairy products for breakfast and did not eat commercially baked goods) had higher health-related quality of life, and lower stress and depressive symptoms, when compared to breakfast skippers.

Together, existing evidence and findings from the current study highlight that breakfast skipping is associated with poorer emotional wellbeing outcomes in child and adolescent populations. The current study supports existing research that demonstrates the link between breakfast skipping and poorer negative emotional wellbeing outcomes. Additionally, results show breakfast skipping was associated with lower positive emotional wellbeing outcomes, and uniquely, our study suggests that the associations between breakfast skipping and emotional wellbeing outcomes for child and adolescent populations may be stronger for positive outcome measures (i.e., life satisfaction, optimism, happiness) compared with negative measures (i.e., sadness, worries). Further, our findings highlight a stepwise association between breakfast skipping and positive wellbeing, whereby the negative relationship between skipping and happiness, life satisfaction, and optimism was highest among students who always skipped breakfast, followed by those who often skipped, and lastly those who sometimes skipped breakfast.

### 4.1. Implications

Child and adolescent mental health and wellbeing is growing in prominence as a key priority for governments internationally. To date, studies focused on breakfast skipping and mental health in child and adolescent populations have rarely included positive wellbeing outcomes. Through the current study, a holistic approach to mental health was taken, in alignment with key principals outlined in the Australian National Children’s Mental Health and Wellbeing Strategy [41]. The strategy emphasizes early mental health and prevention activities in children and adolescents as key principles, and a means of enhancing lifelong wellbeing outcomes. In some other parts of the world, child and adolescent mental health policies maintain focus on mortality prevention, and often face implementation constraints due to deficits in resources and funding [42]. Results from the current study provide important evidence for the relationship between regularly consuming breakfast and positive and negative indicators of emotional wellbeing. Importantly, our findings suggest that child and adolescent wellbeing outcomes may be improved through supports designed to promote regular breakfast consumption.

Historically, Australia has facilitated breakfast consumption promotion and education campaigns via both mass media and targeted school breakfast activities [43]. However, policy and financial investment are lacking for the implementation of school breakfast programs, which have the potential to reduce the prevalence of breakfast skipping [44,45], as seen in countries such as the US and UK [46]. Our findings provide insight into the potential benefits that a range of public health interventions could have for child and adolescent emotional wellbeing outcomes. Potential health promotion campaigns could focus on community education and challenging family health behaviours. Increased investment in school breakfast programs and/or coordinated policy and implementation of student breakfast provision across Australia may help to promote mental health and wellbeing among school-aged students, though further research is necessary to gain a deeper understanding of potential impacts and the mechanisms through which they occur. To inform public health interventions, it will be crucial for future studies to explore the motivators of student breakfast consumption habits. Complex factors such as body image and disordered eating have been found to influence the decision to regularly skip breakfast and are related to child and adolescent mental health [47]. Additionally, variations in culture between communities and families may also contribute to children’s eating behaviours, as well as emotional wellbeing outcomes [48].

### 4.2. Strengths and Limitations

An important strength of this study is that the measurement of emotional wellbeing is aligned with the complete state mental health approach. Happiness, life satisfaction, optimism, sadness, and worries were all measured in this study, utilizing measurement tools with established reliability and validity [29]. We used a population-level, self-report survey of child and adolescent breakfast consumption and wellbeing, increasing the ability to generalize study findings to other child and adolescent populations across Australia and similar countries. Importantly, the self-reported nature of the WEC data collection allows students to voice their experiences of health behaviours and wellbeing outcomes [29,31]. Finally, there is strength in the methodology of our study as it was able to account for a comprehensive set of child and family level confounding variables that influence the relationship between breakfast consumption and emotional wellbeing.

The cross-sectional design of this study limits the ability to draw conclusions regarding temporality. Indeed, longitudinal studies focused on child and adolescent breakfast consumption are lacking [7], and future research should seek to explore effects of breakfast skipping on later outcomes to gain a deeper understanding of long-term effects on child health, development, wellbeing and education outcomes. Whilst a comprehensive set of child- and family-level information, known to influence both breakfast consumption and emotional wellbeing, was used as confounding variables, there may be additional unmeasured confounding factors that could influence the relationship. Due to the restrictions of using secondary data, this study was limited to a selection of measures that were collected by the Department for Education. For example, body mass index, body image, and disordered eating may influence both eating habits and emotional wellbeing, but were unable to be included in the current analysis. Additionally, the study sample was restricted to students attending government schools, excluding students from other education sectors. While the majority of children and adolescents in South Australia attend government schools, these students on average live in communities with a lower socio-economic position compared to students attending non-government schools [49], and so results from this study may over-represent the more socioeconomically disadvantaged students in the state. Future studies investigating breakfast skipping and emotional wellbeing with students from different jurisdictions would further strengthen the depth of knowledge in this space. Finally, our study was not able to incorporate additional detail surrounding breakfast habits (e.g., breakfast content, parental breakfast habits) that may influence the relationship between consumption and wellbeing outcomes. Exploring how breakfast nutritional quality influences child and adolescent emotional wellbeing is also an important avenue for future research [4,42,50]. Additionally, social environments, specifically parental breakfast habits, have been identified as a critical difference between adolescent breakfast skippers versus breakfast eaters, which could be incorporated into future studies.

## 5. Conclusions

Building on limited existing research, this study found negative emotional wellbeing outcomes of sadness and worries were higher for children and adolescents who skipped breakfast every day, compared to their peers who never skipped breakfast. Further, self-reported happiness, optimism, and life satisfaction were higher among students who never skipped breakfast, compared to children and adolescents who skipped breakfast daily. Results suggest the potential for student emotional wellbeing to be supported by interventions designed to promote regular breakfast consumption in child and adolescent populations. Future longitudinal research, as well as research investigating the circumstances surrounding child and adolescent breakfast consumption behaviors is needed to inform the development of effective interventions to increase breakfast consumption.

## Figures and Tables

**Figure 1 nutrients-17-01304-f001:**
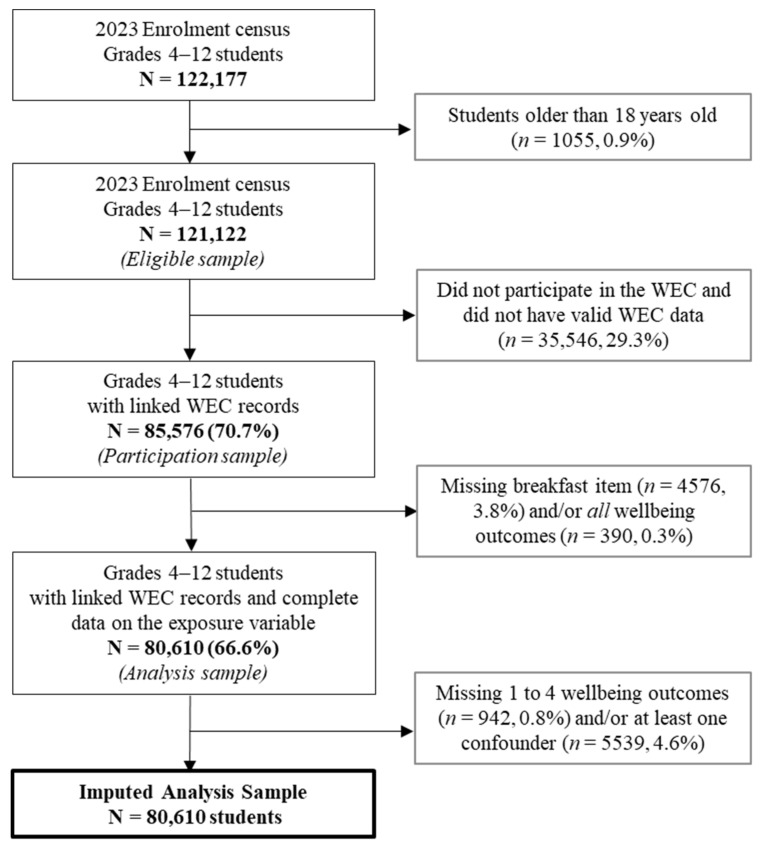
Flow chart for identifying the analysis sample.

**Figure 2 nutrients-17-01304-f002:**
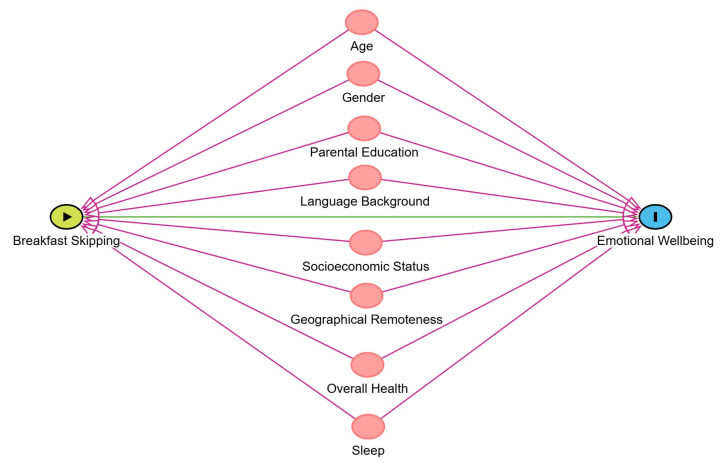
Acyclic graph of confounding factors between breakfast skipping and emotional wellbeing in children and adolescents.

**Figure 3 nutrients-17-01304-f003:**
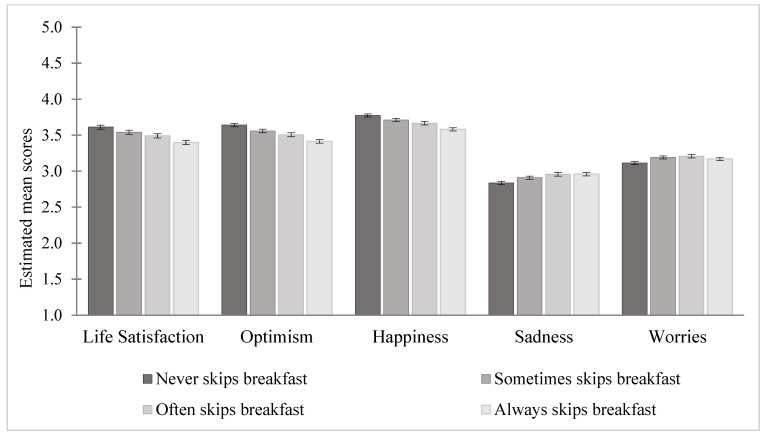
Estimated marginal means on wellbeing indicators by breakfast skipping (*n* = 80,610). Note. The error bars represent 95% confidence intervals.

**Table 1 nutrients-17-01304-t001:** Sociodemographic characteristics by breakfast skipping categories (*n* = 80,610).

	Never Skips	Sometimes Skips	Often Skips	Always Skips
	*n* = 39,711 (49.3%)	*n* = 14,148 (17.6%)	*n* = 17,307 (21.5%)	*n* = 9444 (11.7%)
	*n* (%)/Mean (SD)	*n* (%)/Mean (SD)	*n* (%)/Mean (SD)	*n* (%)/Mean (SD)
Age (years)	12.4 (2.4)	13.1 (2.4)	13.7 (2.4)	14.1 (2.3)
Gender ^1^				
Male	22,516 (56.7%)	7283 (51.5%)	7309 (42.2%)	3926 (41.6%)
Female	17,194 (43.3%)	6865 (48.5%)	9996 (57.8%)	5516 (58.4%)
Language background				
English only	27,911 (70.3%)	10,825 (76.5%)	13,297 (76.8%)	7419 (78.6%)
Non-English	11,800 (29.7%)	3323 (23.5%)	4010 (23.2%)	2025 (21.4%)
Highest parental education level				
Year 12 or below	5484 (13.8%)	2113 (14.9%)	3534 (20.4%)	2304 (24.4%)
Certificate	10,546 (26.6%)	4367 (30.9%)	6105 (35.3%)	3573 (37.8%)
Diploma	5507 (13.9%)	2096 (14.8%)	2543 (14.7%)	1351 (14.3%)
Bachelor’s degree ^2^	18,173 (45.8%)	5573 (39.4%)	5124 (29.6%)	2216 (23.5%)
Socioeconomic position				
1 (Most disadvantaged)	8713 (21.9%)	3185 (22.5%)	5011 (29.0%)	3337 (35.3%)
2	5048 (12.7%)	1986 (14.0%)	2532 (14.6%)	1479 (15.7%)
3	6315 (15.9%)	2221 (15.7%)	2913 (16.8%)	1545 (16.4%)
4	8904 (22.4%)	3174 (22.4%)	3642 (21.0%)	1751 (18.5%)
5 (Most advantaged)	10,732 (27.0%)	3582 (25.3%)	3210 (18.5%)	1331 (14.1%)
Geographical remoteness				
Major cities	29,011 (73.1%)	9923 (70.1%)	11,871 (68.6%)	6424 (68.0%)
Inner regional	5179 (13.0%)	2076 (14.7%)	2618 (15.1%)	1476 (15.6%)
Outer regional	4101 (10.3%)	1604 (11.3%)	2117 (12.2%)	1209 (12.8%)
Remote/very remote	1420 (3.6%)	546 (3.9%)	701 (4.1%)	335 (3.5%)
Overall self-rated health				
Poor/Fair	5259 (13.2%)	3088 (21.8%)	5981 (34.6%)	4537 (48.0%)
Medium	18,755 (47.2%)	7726 (54.6%)	8744 (50.5%)	3685 (39.0%)
Excellent	15,697 (39.5%)	3334 (23.6%)	2583 (14.9%)	1222 (12.9%)
Frequency of a good night sleep				
Never	1581 (4.0%)	757 (5.3%)	1943 (11.2%)	2101 (22.3%)
Sometimes	6614 (16.7%)	4043 (28.6%)	7314 (42.3%)	4332 (45.9%)
Often	17,282 (43.5%)	7159 (50.6%)	5741 (33.2%)	1810 (19.2%)
Always	14,234 (35.8%)	2189 (15.5%)	2309 (13.3%)	1201 (12.7%)

Note. SD = standard deviation; ^1^ gender other than male and female has not been presented due to low numbers (<1%); ^2^ bachelor’s degree or above.

**Table 2 nutrients-17-01304-t002:** Emotional wellbeing outcomes by breakfast skipping categories (*n* = 80,610).

	Never Skips	Sometimes Skips	Often Skips	Always Skips	Total Sample
	*n* = 39,711 (49.3%)	*n* = 14,148 (17.6%)	*n* = 17,307 (21.5%)	*n* = 9444 (11.7%)	*n* = 80,610 (100.0%)
	Mean (SD)	Mean (SD)	Mean (SD)	Mean (SD)	Mean (SD)
Happiness	3.94 (0.68)	3.70 (0.69)	3.48 (0.75)	3.25 (0.88)	3.72 (0.76)
Life Satisfaction	3.82 (0.82)	3.53 (0.84)	3.26 (0.88)	2.99 (0.97)	3.55 (0.91)
Optimism	3.83 (0.81)	3.55 (0.81)	3.29 (0.85)	3.03 (0.95)	3.57 (0.88)
Sadness	2.65 (0.95)	2.91 (0.89)	3.17 (0.91)	3.33 (1.00)	2.89 (0.97)
Worries	2.94 (1.04)	3.21 (0.96)	3.42 (0.95)	3.50 (1.04)	3.15 (1.03)

Note. SD = standard deviation.

**Table 3 nutrients-17-01304-t003:** Results of linear regression analyses examining the effect of breakfast skipping on emotional wellbeing outcomes (*n* = 80,610).

	Unadjusted	Adjusted
	β (95% CI)	β (95% CI)
Happiness		
Never skips	Ref	Ref
Sometimes skips	−0.23 [−0.25, −0.22]	−0.06 [−0.07, −0.05]
Often skips	−0.45 [−0.47, −0.43]	−0.11 [−0.12, −0.09]
Always skips	−0.69 [−0.72, −0.66]	−0.19 [−0.21, −0.17]
Life Satisfaction		
Never skips	Ref	Ref
Sometimes skips	−0.29 [−0.31, −0.27]	−0.07 [−0.09, −0.06]
Often skips	−0.56 [−0.58, −0.53]	−0.12 [−0.14, −0.10]
Always skips	−0.83 [−0.86, −0.80]	−0.22 [−0.24, −0.19]
Optimism		
Never skips	Ref	Ref
Sometimes skips	−0.28 [−0.30, −0.26]	−0.08 [−0.10, −0.07]
Often skips	−0.54 [−0.56, −0.51]	−0.13 [−0.15, −0.12]
Always skips	−0.80 [−0.83, −0.77]	−0.23 [−0.25, −0.21]
Sadness		
Never skips	Ref	Ref
Sometimes skips	0.26 [0.24, 0.28]	0.07 [0.06, 0.09]
Often skips	0.51 [0.50, 0.53]	0.12 [0.11, 0.14]
Always skips	0.67 [0.65, 0.70]	0.12 [0.10, 0.15]
Worries		
Never skips	Ref	Ref
Sometimes skips	0.27 [0.25, 0.29]	0.08 [0.06, 0.09]
Often skips	0.48 [0.46, 0.50]	0.10 [0.08, 0.11]
Always skips	0.56 [0.54, 0.59]	0.05 [0.03, 0.08]

Note: β = unstandardised beta-coefficient. The adjusted regression analysis included the following confounding variables: age, gender, language background, Socio-economic Indexes of Relative Socio-economic Advantage and Disadvantage, highest level of parental education, Accessibility and Remoteness Index of Australia, overall health, and sleep. All analyses were statistically significant *p* < 0.001.

## Data Availability

The datasets presented in this article are not readily available; data access is restricted to researchers who have obtained approval from an accredited research ethics committee to access the datasets. Requests to access the datasets should be directed to the data custodian, the South Australian Department for Education (DfE) website (www.education.sa.gov.au) or via email (education.researchunit@sa.gov.au).

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
