# Peer review of "The Association Between Breakfast Skipping and Positive and Negative Emotional Wellbeing Outcomes for Children and Adolescents in South Australia"

_nutrients, 2025, doi:10.3390/nu17081304_

Round 1
Reviewer 1 Report
Comments and Suggestions for Authors
Dear Authors,
Thank you for your manuscript. This is a well-written paper with clear statistical analyses and a large, impressive sample of South Australian schoolchildren. The manuscript contributes valuable insights into the associations between breakfast skipping and both positive and negative aspects of emotional well-being. Please see my comments below.
Major
Novelty and rationale. The Introduction provides a thorough overview of existing studies investigating the association between breakfast skipping and mental health. However, the novelty of the current study is not clearly articulated. The authors should better emphasize what this study adds to existing knowledge — beyond simply including both positive and negative mental health indicators.
Psychological pathways and theoretical framework. The manuscript would benefit from a more detailed discussion of potential psychological mechanisms linking breakfast skipping and emotional well-being. In particular, issues related to body image, body mass index (BMI), and disordered eating behaviors are not addressed. These are well-established contributors to mental health in adolescents and could be important confounders or mediators in this association.
Structure issues. At the end of the Introduction, the authors begin describing details about the data collection and study measures. These sections belong in the Methods and should be restructured accordingly. Additionally, it is not entirely clear if the authors clearly state the study aim at the end of the Introduction; this should be explicit and easily identifiable.
Lack of important confounders and limitations. BMI is not included in the list of confounders. Given its strong relationship with both breakfast skipping and mental health outcomes, its absence is a notable limitation that should be acknowledged. I recommend that the authors include this point in the limitations section and discuss how the lack of BMI adjustment may impact the interpretation of their findings.
Minor
The discussion could be enriched by acknowledging cultural or socioeconomic factors influencing breakfast habits in more detail.
Consider suggesting practical implications beyond school breakfast programs, such as family-based interventions.
Author Response
Comment 1: Novelty and rationale. The Introduction provides a thorough overview of existing studies investigating the association between breakfast skipping and mental health. However, the novelty of the current study is not clearly articulated. The authors should better emphasize what this study adds to existing knowledge — beyond simply including both positive and negative mental health indicators.
Response 1: Thank you for the suggestion. Additional detail has been added to the introduction to further highlight the strengths of the current study (page 3, line 102-108).
“Furthermore, this study positions children’s voice as critical to the research by allowing students to self-report their own feelings about their wellbeing without relying on others to make assumptions about their mental health, Finaly, by using data captured at a population-level the findings can be generalized to a greater diversity of students and better inform public health efforts to support student wellbeing. By addressing these gaps in the field, the study will allow for a clearer understanding …”
Comment 2: Psychological pathways and theoretical framework. The manuscript would benefit from a more detailed discussion of potential psychological mechanisms linking breakfast skipping and emotional well-being. In particular, issues related to body image, body mass index (BMI), and disordered eating behaviors are not addressed. These are well-established contributors to mental health in adolescents and could be important confounders or mediators in this association.
Response 2: The following has been added to the implications section (page 11, line 392 - 396)
“To inform public health interventions it will be crucial for future studies to explore the motivators of student breakfast consumption habits. Complex factors such as body image and disordered eating have been found to influence the decision to regularly skip breakfast and be related to children and adolescents mental health [44].”
- Wu, X. Y., Yin, W. Q., Sun, H. W., Yang, S. X., Li, X. Y., & Liu, H. Q. (2019). The association between disordered eating and health-related quality of life among children and adolescents: A systematic review of population-based studies. PloS one, 14(10), e0222777. https://doi.org/10.1371/journal.pone.0222777
Comment 3: Structure issues: At the end of the Introduction, the authors begin describing details about the data collection and study measures. These sections belong in the Methods and should be restructured accordingly. Additionally, it is not entirely clear if the authors clearly state the study aim at the end of the Introduction; this should be explicit and easily identifiable.
Response 3: The introduction has been edited to shift details of the study variables to the methods section, putting a greater emphasise the study question.
See changes to the final paragraph of the introduction, with details regarding study measures moved to section 2.3.2 (pg 4-5)
Comment 4: Lack of important confounders and limitations. BMI is not included in the list of confounders. Given its strong relationship with both breakfast skipping and mental health outcomes, its absence is a notable limitation that should be acknowledged. I recommend that the authors include this point in the limitations section and discuss how the lack of BMI adjustment may impact the interpretation of their findings.
Response 4: The following addition has been made to the limitations section of the manuscript (page 12, line 415 - 422).
“Whilst a comprehensive set of child- and family-level information, known to influence both breakfast consumption and emotional wellbeing, was used as confounding variables there may be additional unmeasured confounding factors that could influence the relationship. Due to the restrictions of using secondary data this study was limited to a selection of measures that were collected by the Department for Education. For example, body mass index, body image, and disordered eating may influence both eating habits and emotional wellbeing, but were unable to be included in the current analysis.”
Comment 5: The discussion could be enriched by acknowledging cultural or socioeconomic factors influencing breakfast habits in more detail.
Response 5: Within the current study we have adjusted for socioeconomic factors, including both locality based and parental education socioeconomic influences. This decision was mentioned in our discussion of confounding (page 5, line 203) and supported by our own previous research, in addition to the World Health Organisation and other international studies. However, we acknowledge that this may not sufficiently capture cultural differences within the sample and therefore we have included the following in the implications section of the manuscript (page 12, line 397-398).
“Additionally, variations in culture between communities and families may also contribute to child eating behaviors as well as emotional wellbeing outcomes [45].”
- Scaglioni, S., De Cosmi, V., Ciappolino, V., Parazzini, F., Brambilla, P., & Agostoni, C. (2018). Factors Influencing Children's Eating Behaviours. Nutrients, 10(6), 706. https://doi.org/10.3390/nu10060706
Comment 6: Consider suggesting practical implications beyond school breakfast programs, such as family-based interventions.
Response 6: The implications section has been edited to expand the discussion of potential public health interventions, beyond breakfast programs (page 11, line 385-393)
“Our findings provide insight into the potential benefits that a range of public health interventions could have for child and adolescent emotional wellbeing outcomes. Potential health promotion campaigns could focus on community education and challenging family health behaviors. Increased investment in school breakfast programs and/or coordinated policy and implementation of student breakfast provision across Australia may help to promote mental health and wellbeing among school-aged students, though further research is necessary to gain a deeper understanding of potential impacts and the mechanisms through which they occur.”
Reviewer 2 Report
Comments and Suggestions for Authors
This study offers insights into the relationship between breakfast skipping and emotional wellbeing in children and adolescents.
While the work identifies associations between breakfast skipping and both positive and negative emotional wellbeing outcomes, it does not establish causality. It is possible that pre-existing emotional issues contribute to breakfast skipping rather than the other way around. Longitudinal studies would be necessary to clarify these relationships, as mentioned.
The emotional wellbeing outcomes considered in this study are somewhat narrow. While happiness, optimism, sadness, and worries are important indicators, emotional wellbeing is multifaceted. Other factors such as stress, anxiety, and overall life satisfaction could provide a more comprehensive picture.
The data is drawn from a specific geographical area (South Australia) and may not be representative of broader, more diverse populations. Cultural factors influencing breakfast habits and emotional wellbeing may differ significantly in other regions.
The reliance on self-reported data can introduce bias, as participants may not accurately recall or report their breakfast habits or emotional states. Objective measures or observational studies could strengthen the validity of the findings.
Although the paper mentions adjusted linear regression analyses, it is unclear which confounding variables were controlled for. Factors like socioeconomic status, family dynamics, and academic pressures could influence both breakfast habits and emotional wellbeing.
The conclusion suggests interventions promoting breakfast consumption could enhance emotional wellbeing. However, without a deeper understanding of the underlying reasons for breakfast skipping, such interventions may be too simplistic and fail to address the root causes of the issue.
Author Response
Comment 1: The emotional wellbeing outcomes considered in this study are somewhat narrow. While happiness, optimism, sadness, and worries are important indicators, emotional wellbeing is multifaceted. Other factors such as stress, anxiety, and overall life satisfaction could provide a more comprehensive picture.
Response 1: We acknowledge that there are numerous outcomes that could be considered for measuring child wellbeing. Unfortunately, we are restricted in our current study to the measures collected by the South Australian Department for Education, which have been validated among children aged 8-18 years and feasible for collection within a population census. The limitations section of the manuscript has been updated to include this restriction (page 12, line 418-422).
“Due to the restrictions of using secondary data this study was limited to a selection of measures that were collected by the Department for Education. For example, body mass index, body image, and disordered eating may influence both eating habits and emotional wellbeing outcomes, but were unable to be included in the current analysis.”
Further to the reviewers comment, we were able to include a measure of life satisfaction as an emotional wellbeing outcome in the study (see section 2.3.2, pg 5), and capture children’s “worries”, which focuses on non-clinical levels of anxiety (see section 2.3.2, pg 5). While we acknowledge there are numerous ways to define emotional well-being, we believe the positive (happiness, optimism, life satisfaction) and negative (sadness, worries) emotional wellbeing outcomes that we have used provide a holistic picture of emotional wellbeing, while setting a foundation for future studies to build on our work with using alternative student wellbeing or clinical mental health measures.
Comment 2: The data is drawn from a specific geographical area (South Australia) and may not be representative of broader, more diverse populations. Cultural factors influencing breakfast habits and emotional wellbeing may differ significantly in other regions.
Response 2: Based on a similar comment from another reviewer, we have included the following in the implications section of the manuscript (page 12, line 397-398).
“Additionally, variations in culture between communities and families may also contribute to child health behaviors as well as emotional wellbeing outcomes [45].”
Furthermore, the following has been added to the limitations section of the manuscript (page 12, line 428-430)
“Future studies investigating breakfast skipping and emotional wellbeing with students from different jurisdictions would further strengthen the depth of knowledge in this space.”
Comment 3: The reliance on self-reported data can introduce bias, as participants may not accurately recall or report their breakfast habits or emotional states. Objective measures or observational studies could strengthen the validity of the findings.
Response 3: Whilst we acknowledge recall bias is a hazard of self-report, however in this study we consider it to be a strength. Students have been asked to report on a typical week of breakfast consumption, capturing what they consider their habitual consumption, rather than needing to recall one particular week. Furthermore, allowing students to self-report their own well-being is considered best practice in mental health literature and brings an important and unique perspective to our study, centring students’ own voice and experience in the data (see Coombes et al., 2021).
Coombes, L., Bristowe, K., Ellis-Smith, C., Aworinde, J., Fraser, L. K., Downing, J., Bluebond-Langner, M., Chambers, L., Murtagh, F. E. M., & Harding, R. (2021). Enhancing validity, reliability and participation in self-reported health outcome measurement for children and young people: a systematic review of recall period, response scale format, and administration modality. Quality of life research : an international journal of quality of life aspects of treatment, care and rehabilitation, 30(7), 1803–1832. https://doi.org/10.1007/s11136-021-02814-4
Comment 4: Although the paper mentions adjusted linear regression analyses, it is unclear which confounding variables were controlled for. Factors like socioeconomic status, family dynamics, and academic pressures could influence both breakfast habits and emotional wellbeing.
Response 4: Variables that were adjusted for confounding were included in the footnotes of Table 3, however we have added additional clarification in the statistical analysis section to make this clearer (pg 7, 258-261):
“…after adjustment for confounding variables (age, gender, community socioeconomic status, language background other than English, parent education, level of remoteness, student’s overall health and student’s sleep quality).” Each of these measures are also described in detail within section 2.3.3 (pg 5-6)
Comment 5: The conclusion suggests interventions promoting breakfast consumption could enhance emotional wellbeing. However, without a deeper understanding of the underlying reasons for breakfast skipping, such interventions may be too simplistic and fail to address the root causes of the issue.
Response 5: The following addition has been made to the implications section of the manuscript (page 12, line 393 - 398)
“To inform public health interventions it will be crucial for future studies to explore the motivators of student breakfast consumption habits. Complex factors such as body image and disordered eating have been found to influence the decision to regularly skip breakfast and be related to children and adolescents mental health [44]. Additionally, variations in culture between communities and families may also contribute to child health behaviors as well as emotional wellbeing outcomes [45].”
Round 2
Reviewer 1 Report
Comments and Suggestions for Authors
Dear Authors,
Thank you for the improved version of your manuscript. Congratulations!
Reviewer 2 Report
Comments and Suggestions for Authors
After having seen the major modifications in the manuscript, I believe it can now be published.